# Information Theoretic Causal Effect Quantification

**DOI:** 10.3390/e21100975

**Published:** 2019-10-05

**Authors:** Aleksander Wieczorek, Volker Roth

**Affiliations:** Department of Mathematics and Computer Science, University of Basel, CH-4051 Basel, Switzerland; volker.roth@unibas.ch

**Keywords:** directed information, conditional mutual information, directed mutual information, confounding, causal effect, back-door criterion, average treatment effect, potential outcomes, time series, chain graph

## Abstract

Modelling causal relationships has become popular across various disciplines. Most common frameworks for causality are the Pearlian causal directed acyclic graphs (DAGs) and the Neyman-Rubin potential outcome framework. In this paper, we propose an information theoretic framework for causal effect quantification. To this end, we formulate a two step causal deduction procedure in the Pearl and Rubin frameworks and introduce its equivalent which uses information theoretic terms only. The first step of the procedure consists of ensuring no confounding or finding an adjustment set with directed information. In the second step, the causal effect is quantified. We subsequently unify previous definitions of directed information present in the literature and clarify the confusion surrounding them. We also motivate using chain graphs for directed information in time series and extend our approach to chain graphs. The proposed approach serves as a translation between causality modelling and information theory.

## 1. Introduction

Causality modelling has recently gained popularity in machine learning. Time series, graphical models, deep generative models and many others have been considered in the context of identifying causal relationships. One hopes that by understanding causal mechanisms governing the systems in question, better results in many application areas can be obtained, varying from biomedical [1,2], climate related [3] to information technology (IT) [4], financial [5] and economic [6,7] data. There has also been growing interest in using causal relationships to boost the performance of machine learning models [8].

### 1.1. Overview of Relevant Frameworks of Causality Modelling

Two main approaches to describing causality have been established. One is the Neyman-Rubin causal model or the potential outcomes framework. Its foundational idea is that of two counterfactual statements which are considered (e.g., application of a treatment or lack thereof) along with their effect on some variable of interest (e.g., recession of a disease). This effect (difference between the two counterfactual outcomes) is then measured with the average causal effect. Since for a single data point only one counterfactual is observed, any quantification in the Neyman-Rubin potential outcome model entails a fundamental missing data problem. The other main approach to modelling causal relationships are frameworks based on graphical models, predominantly directed acyclic graphs (DAGs). Such models are akin to Bayesian networks but are imbued with a causal interpretation with the idea of an intervention performed on a node. When a variable is intervened on, any influence of other nodes on it is suppressed and the distribution of the remaining variables is defined as their interventional distribution. The effect of an intervention on a given node is then inferred given the structure of the entire DAG. Such DAGs along with inference rules concerning interventions (called causal calculus) were formalised by Pearl and are referred to as Pearlian graphs.

Regardless of the assumed framework, causal effects (understood as counterfactual outcomes or interventional distributions) can be directly estimated only in the presence of randomised experimental (also referred to as *interventional*) data, meaning that any counterfactual outcome or interventional distribution can be measured. Since this is infeasible in many application areas (e.g., effects of smoking or income cannot be obtained this way), attempts to quantify causal effects with observational data only have evolved into an active field within causal models.

Causal reasoning, within any of the assumed frameworks, can be formulated as one of two fundamental questions. Firstly, one can ask which variables influence which and in what order, that is, which counterfactual (or intervening on which nodes) will have an effect on a particular variable and which other variables have to be taken into account while measuring this effect. This is referred to as *causal induction*, *causal discovery* or *structure learning*, since it corresponds to learning the structure of the arrows of the Pearlian graph in the Pearlian framework. Secondly, once the structure of the causal connections between variables has been learnt or assumed, one can ask how to quantify the causal effect of one variable on another, for example, with the aforementioned average causal effect or the interventional distribution of the effect. This in turn is called *causal deduction*. The former question can be tackled with experimental data or exploiting conditional independence properties of the observable distribution (with algorithms such as PC [9] or IC [10]).

The answer to the latter question with observational data commonly involves a two-step procedure.
First, a set of variables confounding the cause and effect variables is found. Confounding of variables *X* and *Y* by *Z* is a notion that formalises the idea of *Z* causally influencing (directly or not) both *X* and *Y* thus impeding the computation of the direct causal influence of *X* on *Y*. In a Pearlian graph, a set of confounders *Z* can be identified with rules of the causal calculus or with graphical criteria called the back-door criterion and front-door criterion. In the Neyman-Rubin potential outcome framework, a cognate criterion for a set *Z* is called strong ignorability and *Z* is frequently referred to as a set of sufficient covariates.After a set of confounders, or sufficient covariates, *Z* has been identified, the effect of *X* on *Y* is quantified. If such a *Z* exists, this can be done with only observational data. In the Pearlian setting, this amounts to the computation of the interventional distribution of *Y* given the intervention on *X*, which can be shown to be equal to conditioning on or adjusting for *Z* in the observational distribution of X,Y,Z. In the Neyman-Rubin potential outcome framework, the effect of *X* on *Y* is frequently measured with the average causal effect of *X* on *Y*, that is, the difference between expectations for the two potential outcomes. Even though exactly one potential outcome is observed, one can estimate the distribution of the missing one with observational data if one conditions on *Z*.

Thus, one first identifies the set of confounders and then uses them to draw conclusions about causal effects from observational data.

### 1.2. Causality Modelling and Information Theory

The goal of this paper is to formulate a comprehensive description of causal deduction as described above in terms of information theory. In particular, we relate to the most common frameworks of causality modelling sketched in Section 1.1 and provide a novel, rigorous translation of the most common causal deduction methods into the language of information theory.

Previous approaches to express causal concept with information theory were based on the notion of directed information. These approaches, however, were either limited to adjusting the concept of *Granger causality* to time series (which lead to a number of inconsistent definitions of directed information for time series) or only amounted to the first step of the causal deduction procedure. Different approaches were also based on different definitions of directed information for time series and general DAGs. In the following, we clearly motivate *directed information* as the measure of *no confounding* and conditional *mutual information* as the measure that quantifies *causal effect* in a unconfounded setting (i.e., where confounding variables have been adjusted for). We also unify the definitions of directed information for general DAGs and time series and extend the definition to chain graphs which makes it possible to introduce a unique definition of directed information for time series. Finally, we respond to criticisms of directed information that were formulated by different authors as intuition-violating counterexamples and show how our approach allows for a comprehensive causal interpretation of directed information along with regular mutual information.

### 1.3. Related Work on Directed Information and Its History

Directed information has been defined differently by various authors. The main discrepancies stem from the level of definition generality (for two time series [11], multiple time series [12,13], generalised structures [14] and DAGs [15]) and from treatment of instantaneous time points in time series [13]. The former has led to a chain of definitions being generalisations of one another while the latter produced inconsistent definitions for time series. We propose an approach subsuming the different definitions: it is based on extending the most general definition to chain graphs in Section 3.

Directed information was originally introduced as a measure for feedback in discrete memoryless channels [11,16]. It was subsequently imbued with a causal interpretation by noting its similarity to the concept of Granger causality [17,18] between two time series: time series $T_{1}$ Granger-causes $T_{2}$ if the past of $T_{1}$ provides more information about the present of $T_{2}$ than the past of $T_{2}$ does. This resulted in two strategies for formalising directed information: one can assume the instantaneous time points of $T_{1}$ to be a part of the past and condition on them [11,19,20,21] or not [13,16,22,23]. Extensions to account for side information [19] and stochastic processes in continuous time [24] have also been put forward.

The original definition of directed information for discrete memoryless channels was subsequently extended to any time ordering with directed stochastic kernels [14]. This definition, in turn, was a special case of the definition introduced by Raginsky [15]: directed information as KL divergence of interventional and observational distributions. It was shown in the same paper that the conditional version of directed information being zero is equal to the backdoor criterion [25] for no confounding.

Comparing observational and interventional distributions in time series was also considered [12,26,27] and resulted in conditional independence formulation of causality equivalent to directed information [13], yet it did not refer to the necessary first step of causal deduction with observational data, which is deconfounding.

Directed information as a measure of strength of causal effect was criticised for vanishing in the presence of direct causal effect in the underlying Pearlian DAG [28,29] as well as for failing to detect the direction of the causal effect [30]. This critique correctly states that directed information alone is not a proper measure of the strength of causal effect, nevertheless it is rendered moot when one correctly interprets directed information as a measure of no confounding (we refer to it in Section 4).

Recently, directed information has been also applied to areas as diverse as learning hierarchical policies in reinforcement learning [31], modelling privacy loss in cloud-based control [32,33], learning polytrees for stock market data [34], submodular optimisation [35], EEG activity description [36], financial data exploration [37,38] and analysis of spiking neural networks [39]. New methods of directed information estimation [40] as well as generalisations to Polish spaces [41] have been proposed. All of this work, however, treats directed information as a measure of causality strength only and ignores its correct interpretation as measure of no confounding (i.e., the first step in the causal discovery procedure).

### 1.4. Related Work on Graphical Models for Causality

Causal relationships are frequently represented with graphical models, that is, sets of random variables depicted as nodes and relationships between them represented by different types of edges. The basic goal of such models is to encode *dependence structures* of the underlying probability distribution with graph theoretical criteria such as d-separation [42]. When used in the context of causality modelling, one also makes sure that the connections between nodes have a causal interpretation, such as the *data generating process* of the underlying distribution for arrows [25].

A simple graphical model used for causality is the Pearlian DAG encoding both conditional independence relations and the causal data generating process with arrows. Capturing additional information about the dependence structure with the graph theoretic criterion was the motivation for more elaborate graphical models [43]. Completed Partially Directed Acyclic Graphs [9] allow both directed and undirected edges and describe equivalence classes of DAGs which encode the same conditional independence relations with d-separation. Ancestral graphs [44] extend the set of edges with bi-directed edges and allow one to model hidden and selection variables by assuring closure with respect to marginalisation and conditioning. Further extensions of ancestral graphs include maximal ancestral graphs and partial ancestral graphs [45], the latter being the output of popular structure learning algorithms such as FCI [9].

Another motivation for extending the simple DAG model stems from considering the data generating process rather than trying to encode more information about conditional independence relations alone. From the point of view of the data generating process, a DAG describes a set of relationships, where each variable is generated from the set of its parents and external noise [46,47]. *Chain graphs* are an extension of DAGs in which undirected edges between nodes are allowed as long as no semi-directed cycles (cycles with directed and undirected edges) emerge [48,49]. The corresponding data generating process consists of two levels. First, as in DAGs, every set of nodes connected with undirected edges (called *chain component*) depends on the set of all of its members’ parents. Secondly, within every chain component, every node depends on all the other nodes without any specified direction of the dependence (which can be modelled as Gibbs sampling of the nodes in the chain component until the pdfs of the nodes reach an equilibrium). This interpretation of chain graphs’ data generating process and Markov properties was proposed by Lauritzen and Wermuth [48,50,51] and later used as a basis for modelling causal relationships [52,53]. We build on this interpretation of chain graphs in Section 3.

Alternative ways of encoding conditional independence relations in chain graphs have been put forward [54] and [55]. Both have subsequently been extended to account for up to two edges between nodes and to exclude only directed cycles (instead of semi-directed ones) [56,57]. The resulting graphical models are called acyclic directed mixed graphs (ADMGs) [58]. Factorisation criteria along with corresponding algorithms have also been considered for both chain graphs and ADMGs [59,60].

### 1.5. Paper Contributions

In this paper, we make the following contributions:we formulate a two step procedure for causal deduction in two most widely known frameworks of causality modelling and show that the proposed information theoretic causal effect quantification is equivalent to it,we relate to various definitions of directed information and unify them within our approach,we clear some of the confusion persistent in previous attempts to information theoretic causality modelling.

The remainder of this paper is structured as follows. Section 2 introduces our method of causal deduction with information theoretic terms. Subsequently, we explain the existing differences between definitions of directed information, motivate chain graphs as a unifying structure and define directed information for chain graphs in Section 3. We relate to the critique of directed information in Section 4. We conclude with final remarks and an outline of future work in Section 5.

## 2. Proposed Method for Causal Effect Identification

In this section, we formalise the two-step causal deduction procedure with information theoretic terms outlined in Section 1.1. Recall that the two steps for quantifying the causal effect of one variable on another are:
S.1Make sure that the variables are not confounded or find a set of variables confounding them.S.2Use the set found in S.1 (if it exists) to quantify the causal effect.

We elaborate on Step S.1 in the existing frameworks of causal deduction in Section 2.1.1 and in the proposed information theoretic framework in Section 2.2.1. Similarly, Step S.2 is described in Section 2.1.2 and Section 2.2.2, respectively. First, we formally define the necessary concepts from the Pearlian and Neyman-Rubin potential outcome frameworks in Section 2.1.1 and Section 2.1.2 and from information theory in Section 2.1.3.

### 2.1. Notation and Model Set-Up

Graphical models, in particular DAGs, are often employed for modelling causal relationships. Pearlian DAGs [25,61,62,63] represent both direct causal relationships between variables (expressed as arrows) and factorisation of the joint probability distribution of the variables (encoded as conditional independence relations).

A Pearlian DAG G=(V,E) with $V=\{X_{1},X_{2},\cdots ,X_{n}\}$ encodes conditional independence relations with d-separation (For the definition and examples of d-separation in DAGs, see textbooks by Lauritzen [42] or by Pearl [25], Chapters 1.2.3 and 11.1.2). This means that any pair of sets of variables in V d-separated by *Z* is conditionally independent given *Z*. The following probability factorisation and data generating process are assumed for a Pearlian DAG ([25], Chapter 3.2.1):(1)$$P\left(X_{1},X_{2},\cdots ,X_{n}\right)=\prod_{i}P\left(X_{i}|pa\left(X_{i}\right)\right)$$
(2)$$X_{i}=f_{i}\left(pa\left(X_{i}\right),U_{i}\right),$$
where $pa(X_{i})$ stands for the set of direct parents of $X_{i}$ and $U_{i}$ are exogenous noise variables. If $U_{i}$ are pairwise independent and each is independent of non-descendants of $X_{i}$, then the corresponding Pearlian DAG is called Markovian. If the joint distribution $P(U_{1},U_{2},\cdots ,U_{n})$ permits correlations between exogenous variables (which can be used, for example, to represent unmeasured common causes for elements of V), the model is called semi-Markovian ([25], Chapter 3.2.1). The general information theoretic language for causality proposed in this paper remains valid for semi-Markovian and non-Markovian models, but we will confine ourselves to the Markov case in the current paper, since it suffices for describing the basic causal concepts such as the back-door criterion and the average causal effect.

The causal meaning of a Pearlian DAG is formalised with the idea of an intervention: intervening on a variable or a set of variables means setting it to a preselected value and suppressing the influence of other variables on it. This results in the interventional distribution defined formally as ([25], Chapter 3.2.3):(3)$$P\left(X_{1},X_{2},\cdots ,X_{n}|\mathrm{do}\left(X_{i}=x_{i}\right)\right)=\left\{\begin{array}{cc} \prod_{j\neq i}P\left(X_{j}|pa\left(X_{j}\right)\right) & \mathrm{if}\;X_{i}=x_{i}\\ 0 & \mathrm{if}\;X_{i}\neq x_{i}. \end{array}\right.$$ This definition formalises the motivating idea of an intervention by leaving out the term $P(X_{i}|pa\left(X_{i}\right))$ from the product. We will denote $\mathrm{do}\left(X_{i}\right):=\mathrm{do}\left(X_{i}=x_{i}\right)$ whenever it does not lead to confusion. Examples of Pearlian DAGs with interventions are given in Figure 1.

The assumed functional characteristic of each child-parent relationship as defined in the data generating process of a Markovian Pearlian DAG (Equation (Equation 2)) encodes the same conditional independence relationships as the standard factorisation in Equation (Equation 1) [46]. Moreover, one can show that the *Causal Markov Condition* holds for a Markovian Pearlian DAG ([47], Theorem 1): the distribution defined by Equation (Equation 2) factorises according to Equation (Equation 1). Finally, the functional characteristic of all $f_{i}$ along with its equivalence to the factorisation according to Equation (Equation 2) and the definition of intervention make it possible to formalise the concept of *modularity* [61] of Pearlian DAGs: for any node X∈V, its conditional distribution given its parents does not depend on interventions on any other nodes in V. When discussing Pearlian DAGs, we will also assume *positivity*: for any X∈V and a set Z⊂V of non-descendants of *X*, P(X=x|Z)>0 with probability 1 for every *x*, that is, none of the modelled events have probability 0. Note that in the light of the above discussion, Pearlian graphs can be interpreted both as Bayesian networks imbued with a causal meaning and as structural equation models (Markovian models with non-parametric $f_{i}$ in Equation (Equation 2)).

The counterpart of the intervention in the Neyman-Rubin causal model are the potential outcomes of a treatment. In the Neyman-Rubin causal model, potential outcomes Y(0) and Y(1) corresponding to a binary treatment variable *X* are equivalent to the interventional distributions of P(Y|do(X=0)) and P(Y|do(X=1)) [64,65,66,67]. Formally, for X,Y∈V and X={0,1} being a binary variable, potential outcomes Y(0) and Y(1) are equal to the interventional distributions of P(Y|do(X=0)) and P(Y|do(X=1)) and variables *X* and *Y* in the potential outcomes model can be modelled as nodes in a Pearlian DAG [25].

Throughout the rest of this paper we will assume G=(V,E) to be a Pearlian DAG as described above.

#### 2.1.1. Controlling Confounding Bias

The interventional distribution can be computed directly whenever arbitrary interventions in the Pearlian DAG can be performed and measured. This corresponds to randomised treatment assignment in the Neyman-Rubin potential outcome framework (e.g., assigning patients randomly to treatment and control groups such that the assignment does not depend on any other variables in the model). The goal of the first step in the procedure of quantification of causal effects (S.1 in Section 2) is to establish if and how it is possible to circumvent the necessity of measuring the interventional distribution or performing a randomised experiment. This is done by searching for a set of variables which make it possible to express the interventional distribution with observational distributions only.

Given the Pearlian DAG G with observational data only (i.e., a sample from a subset of the nodes of the Pearlian DAG), one can specify conditions under which interventional distribution P(Y|do(X)) can be derived [25,68]. If all parents of *X* are measured, it can be shown that Equation (Equation 3) can be transformed to the following form (*adjusting for direct causes*, that is, parents of *X* in G) ([25], Theorem 3.2.2):(4)$$P\left(Y|\mathrm{do}\left(X\right)\right)=\sum_{X^{\prime }\in pa\left(X\right)}P\left(Y|X,X^{\prime }\right)P\left(X^{\prime }\right).$$

The procedure of Equation (Equation 4), that is, conditioning on a set of variables and then averaging by the probability of this set is referred to as *adjusting* and the said set is called the *adjustment set*. This leads to the following general definition.

**Definition** **1**(Adjustment set [69])**.**
*In a Pearlian DAG G=(V,E), for pairwise disjoint X,Y,Z⊆V, Z is an adjustment set relative to the ordered pair (X,Y) if and only if:*
(5)$$P\left(Y|do\left(X\right)\right)=\sum_{Z^{\prime }\in Z}P\left(Y|X,Z^{\prime }\right)P\left(Z^{\prime }\right)=E_{Z}\left[Y|X,Z\right].$$

The point of adjusting is to remove spurious correlations between *X* and *Y* while not introducing new ones. In this light, controlling confounding bias amounts to finding a set *Z* such that, upon adjusting for *Z*, P(Y|do(X)) can be computed from observational data. Equation (Equation 4) shows that the set of parents of *X* is such a set. This result has been generalised thus making it possible to find adjustment sets also in the case where not all variables in the Pearlian DAG are measured. Such adjustment sets must fulfil the back-door criterion. The generalisation is thus called the back-door adjustment.

**Definition** **2**(Back-door criterion, [25], Definition 3.3.1)**.**
*A set of variables Z⊆V satisfies the back-door criterion relative to an ordered pair of variables (X,Y) if it fulfils the two following conditions:*
no node in Z is a descendant of X and,Z blocks (d-separates) all paths between X and Y that contain an arrow into X.
*Z⊆V satisfies the back-door criterion relative to a pair of disjoint subsets of V, (X,Y) if it satisfies the back-door criterion relative to any pair (X,Y) with X∈X, Y∈Y.*

Note that the first condition in Definition 2 is equivalent to *Z* being a set of post-treatment variables or covariates, that is, variables not affected by treatment in the Neyman-Rubin potential outcomes model [64,70].

**Theorem** **1**(Back-door adjustment, [25], Theorem 3.3.2)**.**
*Let X,Y,Z⊆V be disjoint. If Z satisfies the back-door criterion relative to the pair (X,Y), then it is an adjustment set relative to this pair.*

Examples of adjustments sets corresponding to adjusting for direct causes and the back-door criterion are presented in Figure 2.

The observation that adjusting for a set of variables means the removal of spurious correlations without introducing new ones leads to the following definition of no confounding [25,68]:

**Definition** **3**(No confounding, [25], Definition 6.2.1)**.**
*In a Pearlian DAG G=(V,E), an ordered pair (X,Y) with X,Y⊂V, X∩Y=∅ is not confounded if and only if P(Y=y|do(X=x))=P(Y=y|X=x) for all x,y in their respective domains.*

In the context of the Neyman-Rubin potential outcome model, one often deals with confounding by assuming strong ignorability given *Z* [70]: {Y(0),Y(1)}⫫X|Z. It can be shown that strong ignorability implies that *Z* satisfies the back-door criterion by constructing an appropriate Pearlian DAG ([25], Chapter 11.3.2).

#### 2.1.2. Quantifying Causal Effects

In an *unconfounded* setting, that is, after all spurious correlations between cause and effect have been adjusted for (in the cases where it is possible), one can proceed to quantify the strength of the remaining causal effect (i.e., Step S.2 in Section 2). In a Pearlian DAG, the interventional distribution P(Y|do(X)) describes the causal effect of *X* on *Y*. Note that, as described in Section 2.1.1, this distribution can be computed from observational data when an appropriate adjustment set of variables has been measured, be it all direct ancestors of *X* or variables satisfying the back-door criterion. In the Neyman-Rubin potential outcome framework, one of the most common measures of causal strength for binary treatments is the average causal effect, also referred to as average treatment effect [67,71]:

**Definition** **4**(Average Causal Effect [71])**.**
*Let X be a binary treatment variable and Y(1) and Y(0) stand for potential outcomes corresponding to the counterfactuals X=1 and X=0, respectively. Define:*
(6)ACE(X,Y)=E[Y(1)−Y(0)]=E[Y|do(X=1)−Y|do(X=0)].

An equivalent of ACE restricted to a subspace of the population with a given value of a certain variable *Z* which is a non-descendent of the treatment variable *X* can be defined [68,72,73].

**Definition** **5**(Specific Causal Effect, [72], Definition 9.1)**.**
*Let X be a binary treatment variable and Y(1) and Y(0) stand for potential outcomes corresponding to the counterfactuals X=1 and X=0, respectively. Let Z⊆V be a set of non-descendants of X. Define:*
(7)SCE(X,Y)=E[Y|do(X=1),Z=z−Y|do(X=0),Z=z].

The Specific Causal Effect can be thought of as ACE conditional on a particular value of Z=z (also defined as Conditional Average Causal Effect [74,75]) with the additional requirement that *Z* is a non-descendant (or a set of non-descendants) of *X* in the underlying Pearlian DAG.

Clearly, ACE is a function of two interventional distributions P(Y|do(X=1)) and P(Y|do(X=0)). In general, ACE requires the observation of both potential outcomes. In can be shown however that, when a set of variables *Z* satisfies the back-door criterion with respect to *X* and {Y(0),Y(1)} or strong ignorability is assumed, ACE is estimable from observational data [25,68,73]:(8)$$\begin{array}{cc} ACE(X,Y) & =E_{P(Z)}\left[E[Y(1)|Z]-E[Y(0)|Z]\right]\\ \; & =E_{P(Z)}\left[E[Y|\mathrm{do}(X=1),Z]-E[Y|\mathrm{do}(X=0),Z]\right]\\ \; & =E_{P(Z)}\left[E[Y|X=1,Z]-E[Y|X=0,Z]\right]. \end{array}$$

This corresponds to averaging over the SCE given all the possible values of *Z* or over the conditional average causal effect and mirrors the adjustment formula from Definition 1 and Theorem 1.

Despite its simplicity and limitation to the binary case ACE remains one of the most popular measures of causal effect because of its interpretability (for example in the medical setting where it quantifies the effect of a particular treatment strategy).

#### 2.1.3. Information Theory and Directed Information

We now provide definitions of the necessary concepts from information theory as well as the general definition of directed information. Let G=(V,E) be a Pearlian DAG and assume X,Y⊆V.

Define the *Kullback-Leibler divergence* between two (discrete or continuous) probability distributions *P* and *Q* as $D_{KL}\left(P\left(X\right)\;||\;Q\left(X\right)\right)=E_{P(X)}\log\frac{P(X)}{Q(X)}$ and the *conditional Kullback-Leibler divergence* as $D_{KL}\left(P\left(Y|X\right)\;||\;Q\left(Y|X\right)\;|\;P\left(X\right)\right)=E_{P(X,Y)}\log\frac{P(Y|X)}{Q(Y|X)}$

The *mutual information* between *X* and *Y* is then defined as $I\left(X;Y\right)=D_{KL}\left(P\left(X,Y\right)\;||\;P\left(X\right)P\left(Y\right)\right)$ and the *conditional mutual information* given *Z* as $I\left(X;Y|Z\right)=D_{KL}\left(P\left(X,Y,Z\right)\;||\;P\left(X|Z\right)P\left(Y|Z\right)P\left(Z\right)\right)$.

Let $H\left[P(X)\right]=-E_{P(X)}\left[\log P(X)\right]$ denote *entropy* for discrete and *differential entropy* for continuous *X*. Analogously, $H\left[P(X|Y)\right]=-E_{P(X,Y)}\left[\log P(X|Y)\right]$ denotes *conditional entropy* for discrete and *conditional differential entropy* for continuous *X* and *Y*. For discrete variables, define additionally $H\left[P(X|Y=y)\right]=-E_{P(X|Y=y)}\left[\log P(X|Y=y)\right]$ so that the following holds: $H\left[P(X|Y)\right]=E_{P(Y)}\left[H\left[P(X|Y=y)\right]\right]=-E_{P(X,Y)}\left[\log P(X|Y)\right]$.

As pointed out in Section 1.3, several definitions of directed information have been proposed in the literature. We adopt the definition of directed information given in [15]. In Section 3 we show that this definition subsumes other definitions for time series.

**Definition** **6**(Directed Information [15])**.**
*Let X,Y⊆V be disjoint.*
(9)$$I\left(X\to Y\right)=D_{KL}\left(P(X|Y)\;||\;P(X|do(Y))\;|\;P(Y)\right)=E_{P(X,Y)}\log\frac{P(X|Y)}{P(X|do(Y))}$$

One might also consider interventional distribution with conditioning on a set of passive observations. This leads to the definition of conditional directed information [15] for three disjoint sets X,Y,Z⊆V.

**Definition** **7**(Conditional Directed Information [15])**.**
*Let X,Y,Z⊆V be pairwise disjoint.*
(10)$$I\left(X\to Y|Z\right)=D_{KL}\left(P(X|Y,Z)\;||\;P(X|do(Y),Z)\;|\;P(Y,Z)\right)=E_{P(X,Y,Z)}\log\frac{P(X|Y,Z)}{P(X|do(Y),Z)}$$
*Note that the expression P(X|do(Y),Z) means conditioning on Z in the interventional distribution P(X|do(Y)) as defined in Equation (Equation 3) (i.e., the intervention do(Y) is performed before conditioning on Z). In particular,*
(11)$$P\left(X|do\left(Y\right),Z\right)=\frac{P(X,Z|do(Y))}{P(Z|do(Y))}.$$

Thus, conditional directed information compares the effect of conditioning on *Z* in two distributions: observational P(X|Y) and interventional P(X|do(Y)).

### 2.2. Causal Deduction with Information Theory

We now proceed to lay out the two-step procedure for *information theoretic* causal effect quantification. It consists of ensuring that the two sets of random variables between which the causal effect is to be identified are not confounded (possibly given an adjustment set) and subsequently quantifying the causal effect. The former step Step S.1 in Section 2 is achieved with (conditional) directed information; the latter (Step S.2 in Section 2) with (conditional) mutual information.

#### 2.2.1. Controlling Confounding Bias with (Conditional) Directed Information

In the first step of the information theoretic causal effect quantification procedure one checks whether the two variables of interest are not confounded and if they are, whether any set *Z* can serve as adjustment set. It is straightforward to note that the definition of directed information I(X→Y) provided in Definition 6 is equivalent to the criterion for no confounding between (Y,X) (3). This is formalised in 1.

**Proposition** **1.**
*An ordered pair (X,Y) with X,Y⊆V, X∩Y=∅ is not confounded if and only if I(Y→X)=0.*


The extension of this basic result to the case of adjusting for confounding bias with the back-door criterion was formulated in [15]:

**Proposition** **2**(Theorem 1 in [15])**.**
*Let Z⊂V be a set of non-descendants of X and let X∩Y=∅. Then:*
*Z is an adjustment set for the pair (X,Y) if and only if I(Y→X|Z)=0.*


Propositions 1 and 2 formalise the interpretation of directed information: if the (conditional) directed information from *Y* to *X* vanishes, the causal effect of *X* on *Y* is identifiable with observational data, possibly after adjusting for the conditioning set *Z*. If directed information is greater than 0, performing an intervention on *X* has influenced the distribution of *Y*, hence the difference must stem from the connections between *X* and *Y* in V, which were destroyed while intervening on *X* (such connections correspond to *Z* satisfying the back-door criterion). Note that for the identification of the causal effect X→Y, the ’inverse’ directed information I(Y→X) must vanish.

The interpretation of directed information as a measure of no confounding explains the misunderstandings in situations where directed information (or Granger causality, transfer entropy) is used to quantify direct causal influence. We relate to such ’counterexamples’ in Section 4.

#### 2.2.2. Quantifying the Causal Effect with (Conditional) Mutual Information

As shown in Section 2.2.1, I(Y→X)=0 implies that the causal effect of *X* on *Y* can be identified with observational data, for example, according to (Theorem 1 and Equation (Equation 8)). We now show that in this unconfounded setting, (conditional) mutual information captures the causal effect in a manner analogous to the average causal effect.

Quantifying the causal effect of an intervention with an interpretable value requires proposing a meaningful functional summarising the difference between two (or more) distributions. In the Pearl framework, the causal effect is defined as a function from *X* to P(Y|do(X)), so it captures full distributional information about all possible interventions setting *X* to different values. It therefore represents all available information but is difficult to interpret since it consists of a continuous space of probability distributions. In the Neyman-Rubin causal model, the ACE (Definition 4) makes use of the fact that *X* is binary and reduces both resulting distributions to their means.

We prove that by taking the middle ground, one can meaningfully quantify the causal effect with mutual information and conditional mutual information in an unconfounded setting. To this end, we employ the weighted Jensen-Shannon divergence [76,77], which is sensitive to more than just the first moment of a distribution, as a measure of difference between interventional distributions. We then show that SCE and ACE (Definitions 4 and 5) are equivalent to conditional mutual information and mutual information, respectively, when the difference of means is replaced with the Jensen-Shannon divergence.

**Definition** **8**(Weighted Jensen-Shannon Divergence (JSD) [76]). *Let p,q be probability distributions and $\pi_{q},\pi_{r}\in R_{+}\cup \left\{0\right\}$ be weights with $\pi_{q}+\pi_{r}=1$. The weighted Jensen-Shannon divergence (JSD) is defined as:*
(12)$$JSD\left(q\;||\;r\right)=H\left[\pi_{q}q+\pi_{r}r\right]-\pi_{q}H\left[q\right]-\pi_{r}H\left[r\right].$$

Note that JSD is sometimes equivalently defined for $\pi_{q}=\pi_{r}=\frac{1}{2}$ as symmetrised Kullback-Leibler divergence between p,q and $m:=\frac{1}{2}\left(p+q\right)$: $JSD\left(p,q\right)=\frac{1}{2}(D_{KL}\left(p\;||\;m\right)+D_{KL}\left(q\;||\;m)\right)$ [77]. JSD has recently been applied in many machine learning areas such as GANs [78], bootstrapping [79], time series analysis [80] or computer vision [81].

We first show that for two sets of variables which are not confounded, mutual information quantifies the Jensen-Shannon divergence between two interventional distributions corresponding to the application of a treatment and lack thereof (see Appendix A for the proof).

**Proposition** **3**(Quantifying causal effects with mutual information)**.**
*Assume an ordered pair (X,Y) in a Pearlian DAG with X,Y⊆V, X∩Y=∅ and denote the interventional distributions and corresponding weights as follows:*
(13)$$\begin{array}{cc} q=P(Y|\mathrm{do}(X=1)),\; & \pi_{q}=P\left(X=1\right)\\ r=P(Y|\mathrm{do}(X=0)),\; & \pi_{r}=P\left(X=0\right). \end{array}$$
*Then the following holds:*        *if I(Y→X)=0, then I(X;Y)=JSD(r||q).*

We now proceed to show that when two sets of variables are confounded, but a third set satisfying the back-door criterion relative to these two sets exists, Jensen-Shannon divergences between interventional distributions conditioned on a particular value of the third set and averaged over all values of this set are equal to a KL divergence and conditional mutual information, respectively. These divergences are analogous to SCE and ACE with differences of means replaced with JSD.

**Proposition** **4**(Quantifying specific causal effects)**.**
*Assume an ordered pair (X,Y) in a Pearlian DAG with X,Y⊆V, X∩Y=∅ and Z⊂V which satisfies the back-door criterion (Definition 2).*
*Denote the interventional distributions and corresponding weights for a given value of Z=z as follows:*
(14)$$\begin{array}{cc} q_{z}=P\left(Y|do\left(X=1\right),Z=z\right),\; & \pi_{q_{z}}=P\left(X=1|Z=z\right)\\ r_{z}=P\left(Y|do\left(X=0\right),Z=z\right),\; & \pi_{r_{z}}=P\left(X=0|Z=z\right). \end{array}$$
*Then the following holds:*

*if I(Y→X|Z)=0, then $JSD(r_{z}\;||\;q_{z})=D_{KL}\left(P(X,Y|Z=z)||P(X|Z=z)P(Y|Z=z)\right)$.*


The proof is provided in Appendix A. In fact, it suffices that the equivalent of conditional directed information for the particular *z* vanishes: $I\left(X\to Y|Z=z\right):=D_{KL}(P_{X|Y,Z=z}\left|\right|P_{X|do(Y),Z=z}\left|P_{Y,Z=z}\right)=E_{P_{X,Y|Z=z}}\log\frac{P(X|Y,Z=z)}{P(X|do(Y),Z=z)}$.

The following Corollary justifies using conditional mutual information as a measure of causal effect in an unconfounded setting (see Appendix A for the proof).

**Corollary** **1.**
*(Quantifying average causal effects with conditional mutual information) Assume an ordered pair (X,Y) in a Pearlian DAG with X,Y⊆V, X∩Y=∅ and Z⊂V which satisfies the back-door criterion (Definition 2). Denote the interventional distributions and corresponding weights $q_{z}$, $r_{z}$, $\pi_{q_{z}}$, $\pi_{r_{z}}$ as in Equation (Equation 14) in Proposition 4.*

*Then the following holds:*
      *if I(Y→X|Z)=0, then $E_{Z}[JSD(r_{z}\;||\;q_{z})]=I\left(X;Y|Z\right)$.*

Propositions 3 and 4 and Corollary 1 justify using mutual information and conditional mutual information for quantifying causal effects of *X* on *Y* in unconfounded settings (i.e., whenever *X* and *Y* are not confounded or a set *Z* satisfying the back-door criterion exists). This corresponds to Step S.2.

Both directed information and conditional mutual information have been proposed as measures of quantifying causal effects. Both measures have also been criticised for their shortcomings in the ability of capturing these effects [28,29,30,82,83]. In this section we showed that only their combination yields a rigorous framework for causal effect quantification in Pearlian DAGs. Table 1 summarises our approach.

## 3. Unification of Existing Approaches for Time Series

As stated in Section 1.3, before its general formulation given in Definitions 6 and 7, directed information was defined for discrete channels (or, equivalently, time series) [11,16]. This has resulted in the situation where two competing definitions of directed information for time series are in use: with and without incorporating the instantaneous point in the other time series, that is, with or without ’conditioning on the present’. Denote a set of *n* ordered variables in a Pearlian (a time series with *n* time points) DAG as $X^{n}:=X_{1},X_{2},\cdots ,X_{n}$. Formally, directed information between time series $X^{n}$ and $Y^{n}$ was defined as:(15)$$I\left(X^{n}\to Y^{n}\right)=\sum_{i=1}^{n}I\left(X^{i};Y_{i}|Y^{i-1}\right)$$ by Massey [11] and adopted in this form by some authors [19,20,21] with the justification that $X^{n}$ and $Y^{n}$ are “synchronised” and $X_{i}$ and $Y_{i}$ “occur at the same time” ([19], Chapter 3.1.1). In parallel, the following definition of directed information was put forward in [13,16,22,23] with the argument that “since the causation is already known [...], it is notationally convenient to use synchronised time” [13]:(16)$$I\left(X^{n}\to Y^{n}\right)=\sum_{i=1}^{n}I\left(X^{i-i};Y_{i}|Y^{i-1}\right).$$

Moreover, definitions on different levels of generality are present varying from two and multiple time series as above to general DAGs as in Definitions 6 and 7. In this section, we show that both discrepancies vanish when one considers the different definitions of directed information as special cases of Definitions 6 and 7. We thus unify various formulations of directed information and conditional directed information into one.

To this end, we first show in Section 3.1 that the two variants of directed information for time series defined in Equations (Equation 15) and (Equation 16) are indeed special cases of Definition 6 for different Pearlian DAGs corresponding to different intuitive assumptions concerning time ordering. We subsequently extend the DAGs with a third, confounding, time series and derive formulas for conditional directed informations for these DAGs according to Definition 7.

We then relate the reason for the discrepancy between conditioning on the present and lack thereof to the motivation of using chain graphs in causality modelling and introduce chain graphs in Section 3.2.

Note that directed information for a general Pearlian DAG with a given ordering can be obtained by comparing factorisations of the observational and interventional DAGs [15]. Indeed, expressing Definition 6 as
(17)$$I\left(X\to Y\right)=E_{P(X,Y)}\log\frac{P(X|Y)}{P(X|\mathrm{do}(Y))}=E_{P(X,Y)}\log\left[\frac{P(X,Y)}{P(X|\mathrm{do}(Y))P(Y)}\right]$$
results in the observational distribution in the numerator and a product of the interventional distribution and the marginal distribution of the variables intervened upon in the denominator. Factorisations of both distributions can be directly read off the corresponding DAGs. Different forms for directed mutual informations result from the different orderings imposed on the underlying Pearlian DAGs.

### 3.1. Directed Information for Time Series Represented with DAGs.

We now show that the definitions of directed information for time series Equations (Equation 15) and (Equation 16) are special cases of Definition 6. We do this by defining appropriate Pearlian DAGs (corresponding to *full time ordering* and *partial time ordering*) and applying Definitions 6 and 7 as well as factorisations of observational and interventional distributions (Equations (Equation 1) and (Equation 3)) to them.

Consider a Pearlian DAG $G_{1}=\left(V,E\right)$, where |V|=2n and a total order on V is given. This means that $V=(V_{1},V_{2},\cdots ,V_{2n-1},V_{2n})$, with E consisting of all possible arrows pointing to the future, that is, $V_{i}\to V_{j}$ with i<j. Now, define $X^{n}=\left(X_{1},X_{2},\cdots ,X_{n}\right)=\left(V_{1},V_{3},\cdots ,V_{2n-1}\right)$ and $Y^{n}=\left(Y_{1},Y_{2},\cdots ,Y_{n}\right)=\left(V_{2},V_{4},\cdots ,V_{2n}\right)$. DAG $G_{1}$ is depicted in Figure 3. Theorem 2 shows the formula for directed information that follows from applying Definition 6 to $G_{1}$.

**Theorem** **2.**
*In the Pearlian DAG $G_{1}$ directed information from $X^{n}$ to $Y^{n}$ has the following form:*
(18)$$I\left(X^{n}\to Y^{n}\right)=\sum_{i=1}^{n}I\left(X^{i};Y_{i}|Y^{i-1}\right).$$
*In the same DAG $G_{1}$, directed information from $Y^{n}$ to $X^{n}$ has the following form:*
(19)$$I\left(Y^{n}\to X^{n}\right)=\sum_{i=1}^{n}I\left(Y^{i-1};X_{i}|X^{i-1}\right).$$


See Appendix A for the proof. Note that Equation (Equation 18) is indeed equivalent to the directed information defined on time series in [11] (Equation (Equation 15)).

Now consider a Pearlian DAG $G_{2}$ similar to $G_{1}$ ($G_{2}=\left(V,E\right)$, $V=\{X_{1},\cdots ,X_{n},Y_{1},\cdots ,Y_{n}\}$) but with a slight twist. Let now $X^{n}$ and $Y^{n}$ be aligned, that is, indexed at the same time points. Let E again consist of all possible arrows pointing to the future (i.e., all arrows $X_{i}\to X_{j}$, $Y_{i}\to Y_{j}$, $X_{i}\to Y_{j}$, $Y_{i}\to X_{j}$, with i<j). $G_{2}$ is shown in Figure 4a. Applying Definition 6 to $G_{2}$ as well as $G_{2}$ together with a third, confounding, time series (Figure 4b) yields Theorem 3.

**Theorem** **3.**
*In the Pearlian DAG $G_{2}$ directed information from $X^{n}$ to $Y^{n}$ has the following form:*
(20)$$I\left(X^{n}\to Y^{n}\right)=\sum_{i=1}^{n}I\left(X^{i-1};Y_{i}|Y^{i-1}\right).$$
*Conditioning on an aligned time series $Z^{n}$ (see Figure 4b) yields:*
(21)$$I\left(X^{n}\to Y^{n}|Z^{n}\right)=\sum_{i=1}^{n}I\left(X^{i-1};Y_{i}|Y^{i-1},Z^{i-1}\right).$$


See Appendix A for the proof. Analogously to Theorem 2, Equation (Equation 20) is equivalent to the directed information defined on time series in [16] (Equation (Equation 16)).

### 3.2. Factorisations and Interventions in Chain Graphs

In Section 3.1 we showed that two definitions of directed information proposed in the literature are subsumed by Definition 6. These two definitions differ in how they treat events that are supposed to be time-aligned. It is therefore not clear what causal assumptions or hypotheses should be allowed to model such events: if an association is observed between them, can it be explained by a directed arrow in the data generating process in Equation (Equation 2) (and if so, which direction should be assumed), by an unmeasured variable in a semi-Markovian model or can it only be an artefact of the functional form of the other arrows?

Similar considerations have led to the extension of DAGs to chain graphs as graphical models for causality. Potential presence of associations between variables which cannot be attributed to an underlying causal process (e.g., because the direction of causality cannot be established with available measurements, there exists an unmeasured confounding variable or a feed-back mechanism) motivated a causal interpretation of chain graphs [52,53] analogous to the causal interpretation of DAGs introduced in Section 2.1. The said non-causal direct associations are modelled with undirected adges between variables.

A chain graph (CG) H=(V,E) is an extension of DAG in which E can also contain undirected edges and where no semi-directed cycles (i.e., cycles with directed and undirected edges) are allowed. This induces a new relationship between the elements of V, distinct from parenthood: X,Y∈V are called *neighbours* if they are connected by an undirected edge. The set T of connected components (neighbours) of V obtained by removing all directed edges in a chain graph is called the set of chain components. In particular, chain graphs with no undirected edges or where all chain components are singletons are DAGs.

Analogously to Equations (Equation 2) and (Equation 3), the data generating process as well as interventional distribution have been defined for CGs. We follow the approach put forward in [52,53].

The data generating process of a CG is, again, an extension of that of a DAG (Equation (Equation 2)). As mentioned in Section 1.4, it consists of two levels. First, functional relationships of each child-parent pair of chain components are modelled: $\tau =f_{\tau }\left(pa\left(\tau \right),U_{\tau }\right)$ where $pa\left(\tau \right)={⋃}_{X\in \tau }pa\left(X\right)\backslash \tau$. This corresponds to a DAG of all the chain components τ∈T. Secondly, for every chain component τ, a sampling procedure represented by $g_{\tau }$ is performed ([52], Section 6.3):(22)$$\tau =g_{\tau }\left(pa\left(\tau \right)\right).$$ here, $g_{\tau }$ represents the sampling function of the undirected graph τ. It takes all parents of τ as input and for every X∈τ, it samples from its current distribution given pa(τ)∪τ\{X} until reaching an equilibrium.

Just like the data generating process for DAGs motivates the definition of interventional distribution for DAGs (Section 2.1 and Equation (Equation 3)), the same reasoning can be applied to CGs, which leads to the following definition of the interventional distribution in a CG ([52], Section 6.4):(23)$$P\left(X|\mathrm{do}\left(Y\right)\right)=\prod_{\tau \in T}P\left(\tau \backslash Y|pa\left(\tau \right),\tau \cap Y\right).$$

Thus, for every chain component τ that intersects with *Y*, τ∩Y is removed from the factorisation (just like $P(X_{j}|pa\left(X_{j}\right))$ is removed from DAGs in Equation (Equation 3)) but still influences the remainder of the chain component τ by conditioning it. Examples of interventions in chain graphs are presented in Figure 5.

### 3.3. Directed Information for Chain Graphs Representing Aligned Time Series

We now revisit directed information for time series motivated in Section 3.1. We first showed that two versions of directed information present in the literature (Equations (Equation 15) and (Equation 16)) are subsumed by Definition 6 and that the difference in motivations for the two versions is captured by the causal interpretation of chain graphs (Section 3.2). In this section, we propose to model aligned time series explicitly with chain graphs.

To this end, we define chain graph $H_{1}=\left(V,E\right)$, where V and E are as in DAG $G_{2}$ from Theorem 3 and Figure 4a with E extended by additional undirected edges between every pair of $X_{i}$ and $Y_{i}$. Thus, all sets $\left\{X_{i},Y_{i}\right\}$ are chain components. $H_{1}$ is depicted in Figure 6a.

Theorem 4 shows the formula for directed information as well as conditional directed information in chain graphs presented in Figure 6.

**Theorem** **4.**
*In the chain graph $H_{1}$ directed information from $X^{n}$ to $Y^{n}$ has the following form:*
(24)$$I\left(X^{n}\to Y^{n}\right)=\sum_{i=1}^{n}I\left(X^{i-1};Y_{i}|Y^{i-1}\right).$$
*Conditioning on an aligned time series $Z^{n}$ (see Figure 6b) yields:*
(25)$$I\left(X^{n}\to Y^{n}|Z^{n}\right)=\sum_{i=1}^{n}I\left(X^{i-1};Y_{i}|Y^{i-1},Z^{i-1}\right).$$


The proof, again, uses Definitions 6 and 7 and appropriate factorisations of observational and interventional distributions for chain graphs as defined in Equations (Equation 22) and (Equation 23) (see Appendix A).

## 4. Relation to Critique of Previous Information Theoretic Approaches

Directed information has been subject to criticism in the literature [28,29,30,84]. It concerned the time series formulation (as in Equations (Equation 15) and (Equation 16)), also in the form of transfer entropy or information flow. In the latter two forms, only the last term of the sum in Equations (Equation 15) and (Equation 16) is taken as the definition of directed information. All of the critique amounted to constructing examples where directed information fails to mirror intuitions or postulates concerning causal effect quantification. These postulates, however, are usually based on the erroneous assumption that directed information is by definition a measure of causal influence. As we described in Section 2, directed information is a measure of no confounding and constitutes the first step in the two-step procedure of causal effect quantification. In this section, we refer to the most common point of criticism raised in recent literature and show that it becomes irrelevant when one interprets directed information correctly and proceeds according to the information theoretic causal quantification procedure we proposed in Section 2.

Ay and Polani [28] consider a Pearlian DAG depicted in Figure 7. They note that transfer entropy from *X* to *Y* (i.e., directed information $I(X^{n-1}\to Y_{n})$, defined by them as $I(X^{n-1};Y_{n}|Y^{n-1})$) vanishes even though, intuitively, *X* directly influences *Y* (the example is symmetric in *X* and *Y*). Specifically, if one defines all the arrows in Figure 7 as noisy copy operations (i.e., one assumes $X_{i}=Y_{i-1}+\epsilon_{X_{i}}$ and $Y_{i}=X_{i-1}+\epsilon_{Y_{i}}$ with all $\epsilon ∼N(0,\sigma^{2})$ as Equation (Equation 2) in the underlying Pearlian DAG), then $I(X^{n-1}\to Y_{n})$ decreases to 0 as ϵ→0. The same critique was repeated by other authors [29,82]. It can, however, be easily explained with the two step method proposed in Section 2.

According to step S.1 of the causal effect quantification procedure described in Section 2, if one is interested in the causal effect of $X^{n-1}$ on $Y_{n}$, one needs to first analyse the directed information $I(Y_{n}\to X^{n-1})$, since it measures whether the pair $\left(X^{n-1},Y_{n}\right)$ is not confounded. If $I(Y_{n}\to X^{n-1})=0$ in the underlying DAG, one can proceed to quantifying the causal effect with mutual information $I(Y_{n};X^{n-1})$ (Step S.2).

Having established that, note that $I(Y_{n}\to X^{n-1})$ in the DAG from Figure 7 is indeed equal to 0:(26)$$\begin{array}{cc} I(Y_{n}\to X^{n-1}) & =E_{P(X,Y)}\log\frac{P(X^{n-1},Y_{n})}{P\left(Y_{n}|\mathrm{do}\left(X^{n-1}\right)\right)P\left(X^{n-1}\right)}\\ \; & =E_{P(X,Y)}\log\frac{P\left(Y_{n}|X_{n-1}\right)\prod_{i=2}^{n-1}P\left(X_{i}|X_{i-2}\right)\;P\left(X_{1}\right)}{P\left(Y_{n}|X_{n-1}\right)P\left(X_{1}\right)\prod_{i=2}^{n-1}P\left(X_{i}|X_{i-2}\right)}=0. \end{array}$$

Therefore, in order to clarify the criticism of directed information formulated in [28,29,30], it is essential to:use the directed information $I(Y_{n}\to X^{n-1})$ as a measure of no confounding,calculate $I(Y_{n}\to X^{n-1})$ according to the underlying DAG presented in Figure 7 (Equation (Equation 26)).

## 5. Conclusions

In this paper, we have proposed an attempt to bridge the most popular frameworks of causality modelling with information theory. To this end, we described a two step procedure of causal deduction, consisting of identifying confounding variables and subsequently quantifying the causal effect in an unconfounded setting, in each of these frameworks. We then expressed this procedure with infromation theoretic tools. Subsequently, we unified different definitions of directed information and clarified some of the confusion surrounding its causal interpretation. This is relevant since previous approaches to interpreting directed information were largely limited to the setting of time series and erroneously attributed causal effect quantification to directed information.

The full information theoretic description of causal deduction can be of interest to two communities. Firstly, for the statistical and causality community, since it provides a direct translation to the language of information theory, which has made inroads into machine learning recently. Secondly, it allows for the use of information theoretic machine learning models, such as the variational auto-encoder [85,86], deep information bottleneck [87,88], InfoGAN [89], and so forth, for causality modelling and integrating causal deduction in such models. The latter approach has already sparked interest in recent machine learning literature, for example, in the context of using causal relationships to facilitate transfer learning in deep models [8,90], explaining deep generative models and making them more interpretable [91,92] and boosting the performance of deep neural networks [93].

Future work includes elucidating information theoretic equivalents of further causal concepts such as the effect of treatment on the treated, propensity score based methods or double robustness models.

## Figures and Tables

**Figure 1 entropy-21-00975-f001:**
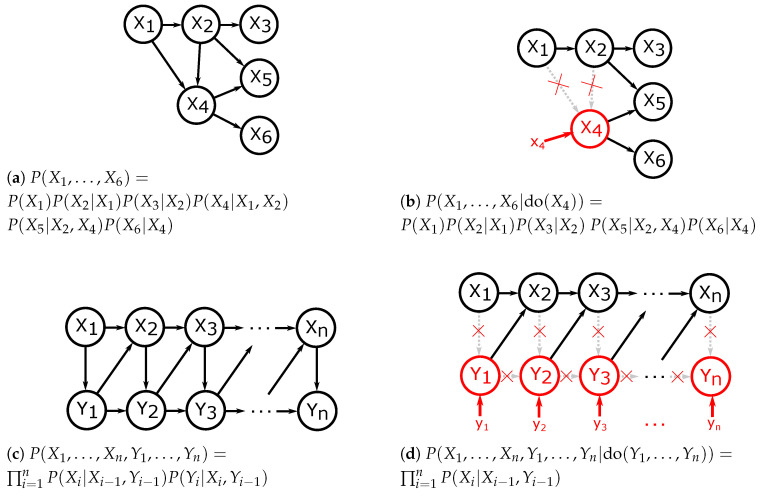
Examples of interventions performed on directed acyclic graphs (DAGs) with resulting probability factorisations. Left: observational distributions and factorisations. Right: interventional distributions and factorisations.

**Figure 2 entropy-21-00975-f002:**
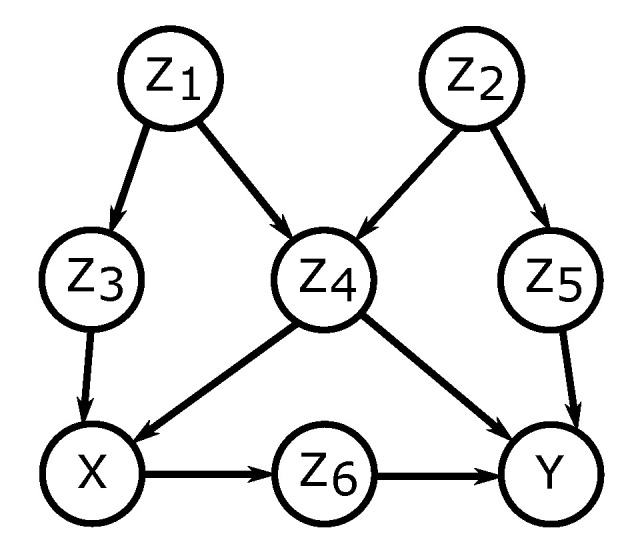
Adjustment sets for (X,Y). Back-door adjustment [25]: $\left\{Z_{3},Z_{4}\right\}$ and $\left\{Z_{4},Z_{5}\right\}$ satisfy the back-door criterion with respect to (X,Y). Only the former corresponds to adjusting for direct causes of *X*.

**Figure 3 entropy-21-00975-f003:**
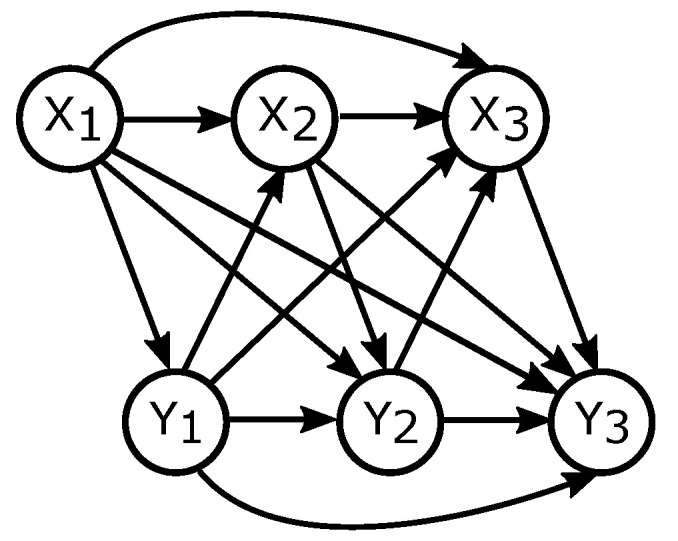
Pearlian DAG $G_{1}$ representing *full ordering* considered in Theorem 2.

**Figure 4 entropy-21-00975-f004:**
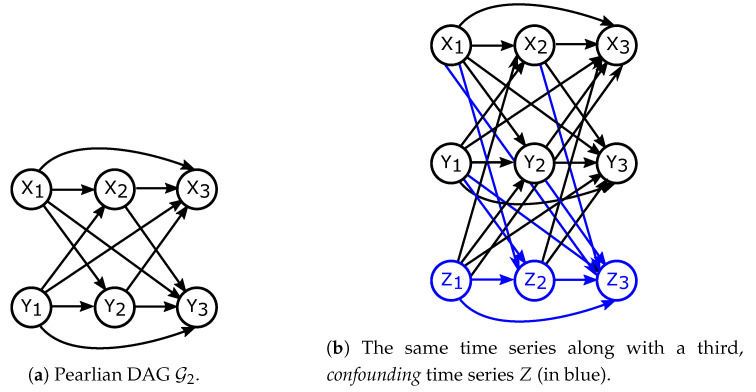
Pearlian DAGs representing *partial ordering* considered in Theorem 3.

**Figure 5 entropy-21-00975-f005:**
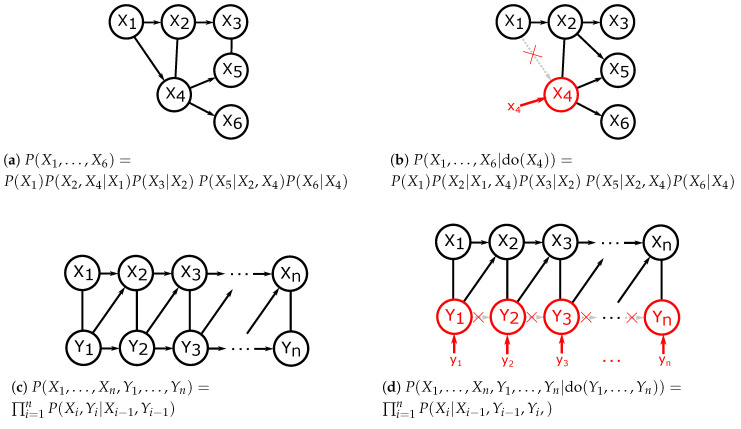
Examples of interventions performed on chain graphs with resulting probability factorisations. Left: observational distributions and factorisations. Right: interventional distributions and factorisations. Note that, as opposed to Figure 1, $\left\{X_{2},X_{4}\right\}$ (Figure 5a,b) and $\left\{X_{i},Y_{i}\right\}$ (Figure 5c,d) form chain components.

**Figure 6 entropy-21-00975-f006:**
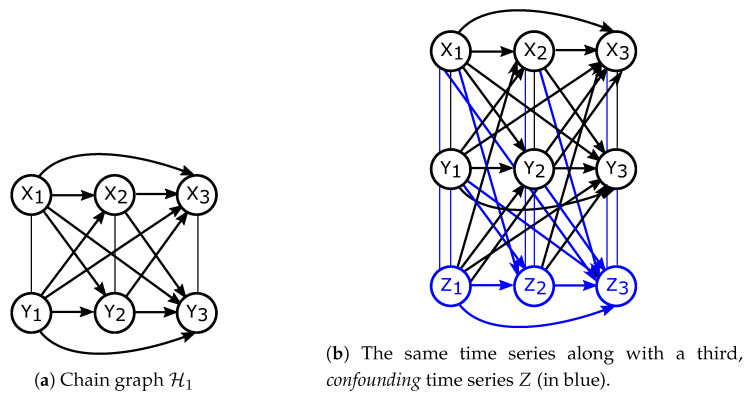
Chain graphs considered in Theorem 4.

**Figure 7 entropy-21-00975-f007:**
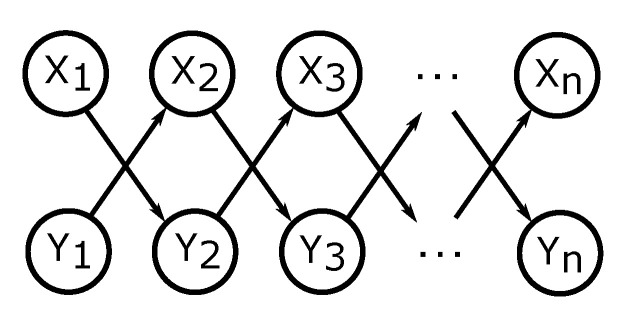
Example of “vanishing directed information” [28,29,82].

**Table 1 entropy-21-00975-t001:** Summary of information theoretic causal effect quantification and comparison of the two steps to the Pearl and Neyman-Rubin potential outcome frameworks.

	Pearlian Framework	Neyman-Rubin Potential Outcome Framework	Information Theoretic Framework
Ensuring no confounding	back-door criterion	strong ignorability	conditional directed information =0
Causal effect quantification	interventional distribution	average causal effect	conditional mutual information

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
