# Peer review of "Information Theoretic Causal Effect Quantification"

_entropy, 2019, doi:10.3390/e21100975_

Round 1
Reviewer 1 Report
Information Theoretic Causal Effect Quantification (entropy-596444)
by Wieczorek and Roth
In this paper, the authors extended the causal effect quantification theory between random variables using Pearlian causal and the Neyman-Rubin frameworks, both under an information theoretic approach. They proposed defined several steps (no confounding, adjustment, causal effect quantification, etc) to then, unified these steps for time series directed information based on chain graphs.
I think that the paper is well written, provides new results, and the results and discussion are well presented.
Only some few specific details must to be studied by the author. In order that authors reply these comments/suggestions (see below), I could recommends the publication of this article. Also, additional references are suggested below for your consideration.
L20: "...and computer sciences [Dourado et al., 2019] data".
L81: " of Granger causality [15,16]".
L106: delete title "2. Related Work", and Subsections 2.1 and 2.2 can be enumerated as 1.3 and 1.4, respectively; as part of Introduction section. Then, move the lines 92-105 at the end of Introduction.
L109: "multiple time series [11,24],".
L127: "[24,25, Jafari-Mamaghani and Tyrcha (2014)]".
L152-153: revise grammar of "more... [41].".
L191: Why some words are marked in bold? (revise in all document).
L215: This is a probability? This is always $>0$.
L251: put "." at the end of this line.
L358(+1) & 372: "Divergence" <-> "Distance".
L363: "... or time series analysis [Contreras-Reyes, 2016]". Also, the JSD correspond to a distance: the triangular inequality is accomplished [Contreras-Reyes, 2016].
Proofs of Propositions 3 and 4 and Corollary 1 can be moved in Appendix as the Theorem's proofs.
L398(+2): "[10,14, Sun and Bollt, 2014]".
L414: put this expression in a whole line.
L503: Please, describe the assumptions for epsilon noise: mean, variance, distribution.
References:
Dourado, J.R., Júnior, J.N.D.O., Maciel, C.D. (2019).
Parallelism Strategies for Big Data Delayed Transfer Entropy Evaluation.
Algorithms 12(9), 190.
Jafari-Mamaghani, M., Tyrcha, J. (2014).
Transfer entropy expressions for a class of non-Gaussian distributions.
Entropy 16(3), 1743-1755.
Contreras-Reyes, J.E. (2016).
Analyzing fish condition factor index through skew-gaussian information theory quantifiers. Fluctuation and Noise Letters 15(02), 1650013.
Sun, J., Bollt, E. M. (2014).
Causation entropy identifies indirect influences, dominance of neighbors and anticipatory couplings. Physica D 267, 49-57.
Author Response
We are truly appreciative to the reviewer for the comments and suggestions for improving our paper. We updated out manuscript accordingly and relate to the points raised by the reviewer below.
L20: "...and computer sciences [Dourado et al., 2019] data".
-> Done: we added the citation.
L81: " of Granger causality [15,16]".
-> We cite Granger causality [15,16] in related work, L116.
L106: delete title "2. Related Work", and Subsections 2.1 and 2.2 can be enumerated as 1.3 and 1.4, respectively; as part of Introduction section. Then, move the lines 92-105 at the end of Introduction.
-> Done.
L109: "multiple time series [11,24],".
-> Done.
L127: "[24,25, Jafari-Mamaghani and Tyrcha (2014)]".
-> Done: we added the additional citation.
L152-153: revise grammar of "more... [41].".
-> Done. It now says “Capturing additional information about the dependence structure with the graph theoretic criterion was the motivation for more elaborate graphical models [43]”
L191: Why some words are marked in bold? (revise in all document).
-> Done.
L215: This is a probability? This is always $>0$.
-> P(X=x|Z)>0 for any x, i.e. the positivity assumption requires that P(X=x|Z)=0 never happens (which means that no events have probability 0). This is a technical requirement for Eq. (5) and Proposition 4 to hold: otherwise, one would have to condition on events with probability 0 while generalising specific causal effects to average causal effects. We added a short clarification in L215 to avoid confusion.
L251: put "." at the end of this line.
-> Done.
L358(+1) & 372: "Divergence" <-> "Distance".
-> We introduce the Jensen Shannon "Divergence" in Definition 8 and always refer to it as "divergence". This is to avoid confusion with the Jensen Shannon "Distance" which is sometimes defined as the square root of the Jensen Shannon "Divergence" from Definition 8 and is a metric.
L363: "... or time series analysis [Contreras-Reyes, 2016]". Also, the JSD correspond to a distance: the triangular inequality is accomplished [Contreras-Reyes, 2016].
-> Done: we added the citation on JSD in time series analysis.
Proofs of Propositions 3 and 4 and Corollary 1 can be moved in Appendix as the Theorem's proofs.
-> Done.
L398(+2): "[10,14, Sun and Bollt, 2014]".
-> We added the citation on causation entropy to the section on the critique of directed information (L488).
L414: put this expression in a whole line.
-> Done.
L503: Please, describe the assumptions for epsilon noise: mean, variance, distribution.
-> Epsilon noise in the example we reference are i.i.d. Gaussian variables with zero mean and sigma^2 variance. We added this description to the manuscript.
Reviewer 2 Report
The paper is informative and well documented and attempts a unification much of causality indicators within an information-theoretic framework. The limitations, though, of the proposed setting, are not clearly indicated. Also, the lack of some guidingexamples makes the paper hard to read for a not deeply mathematically motivated reader. I would strongly suggest that the authors take some time to improve on these remarks although the paper is publishable as-is, but not expected to motivate a wider audience in studying and following it.
Author Response
We would like to thank the reviewer for the comments and for appreciating our work.
We state the limitations of our approach in that we define it as a causal deduction method as opposed to causal induction (line 51, line 75 in Section 1): this means that a (partial) structure, assumed or learnt e.g. with a structure learning algorithm, must be provided. Another limitation might stem from estimating (conditional) mutual informations as in Eqs. (10—13, 16,17), but here any MI estimation technique can be plugged in, varying from assuming a Gaussian copula of the variables to non-parametric approaches such as the Kraskov estimator.